# Neuroprosthetic contact lens enabled sensorimotor system for point-of-care monitoring and feedback of intraocular pressure

Weijia Liu[1], Zhijian Du[1], Zhongyi Duan[1], La Li ⬀[1] ✉ & Guozhen Shen ⬀[1] ✉

The wearable contact lens that continuously monitors intraocular pressure (IOP) facilitates prompt and early-state medical treatments of oculopathies such as glaucoma, postoperative myopia, etc. However, either taking drugs for pre-treatment or delaying the treatment process in the absence of a neural feedback component cannot realize accurate diagnosis or effective treatment. Herein, a neuroprosthetic contact lens enabled sensorimotor system is reported, which consists of a smart contact lens with $Ti_3C_2T_x$ Wheatstone bridge structured IOP strain sensor, a $Ti_3C_2T_x$ temperature sensor and an IOP point-of-care monitoring/display system. The point-of-care IOP monitoring and warning can be realized due to the high sensitivity of 12.52 mV mmHg$^{-1}$ of the neuroprosthetic contact lens. In vivo experiments on rabbit eyes demonstrate the excellent wearability and biocompatibility of the neuroprosthetic contact lens. Further experiments on a living rate in vitro successfully mimic the biological sensorimotor loop. The leg twitching (larger or smaller angles) of the living rat was demonstrated under the command of motor cortex controlled by somatosensory cortex when the IOP is away from the normal range (higher or lower).

Glaucoma, as the second leading cause of irreversible blindness, affects almost 80 million people worldwide in 2023, which is expected to increase to 118 million by 2040[1,2]. 72% of glaucoma patients are detected at late stages, missing the optimal time for treatment[3]. Therefore, finding a way to detect glaucoma earlier and taking preventative action are important[4,5]. In this regard, the increase of intraocular pressure (IOP) is seen as the most obvious premonitory symptom, and hence continuously monitoring IOP can reduce the incidence of blindness caused by glaucoma[6–8]. Additionally, point-of-care IOP monitoring decides no less significant than glaucoma for post-surgical patients undergoing myopia correction (a procedure increasing year by year and expected to reach 5 billion cases by 2050) and eye-overuse group[9,10].

The rise of wearable smart contact lens (SCL) with additional functions beyond vision correction, through the integration of diverse electronic sensors, microprocessors, communication, and display components to provide non-invasive, continuous IOP monitoring, is effective in accurate diagnosis and treatments of oculopathies[11–13]. For example, Kim et al. developed a smart contact lens consisting of a gold nanotube-based strain sensor, a drug delivery system and integrated circuits to simultaneously monitor and control IOP via timolol release[14]. Park and co-workers reported a smart contact lens with a Si strain sensor for the accurate measurements of IOP[15]. However, the lack of the biological feedback function in these reported smart contact lenses cannot mimic natural IOP stimulated-nerve-induced motor activities, which failed to compensate for the limited or neglected nature of the information provided by patients because of the

[1]School of Integrated Circuits and Electronics, Beijing Institute of Technology, 100081 Beijing, China. ✉e-mail: lali@bit.edu.cn; gzshen@bit.edu.cn

individual variations in pain perception and sympathetic responses, leading to a shortfall in providing timely and effective medical treatments[16,17]. Therefore, constructing an association between high IOP and somatosensory cortex is of great significance. Retinomorphic neurons and artificial reflex arcs based on neuromorphic devices have been demonstrated to build the relationship between stimulate and neurons[18–20].

Facing the above-mentioned challenges, we proposed a biocompatible and stable neuroprosthetic contact lens (NCL) through material engineering, device architecture simulation, smart contact lens fabrication, nerve-motor feedback, system integration, and functional verification. The fabricated NCL could convert temperature-corrected IOP information to an electric signal, then transfer it to the somatosensory cortex, and lead to the motor cortex issuing instructions to the sciatic nerve, finally making leg twitching, forming a closed-loop system from IOP signal generating nerve-perception to motor activities. The strain gauges and temperature sensors in the neuroprosthetic contact lens are made of $Ti_3C_2T_x$ MXene materials, which reduce the complexity of the fabrication and improve the compatibility with soft hemispherical substrates. A high sensitivity of $12.52 \, mV \, mmHg^{-1}$ and superior stability (no significant performance degradation after 1 h) have been achieved for the $Ti_3C_2T_x$ MXene-based strain sensor attributed to the introduction of serpentine electrode design, which enables the temperature-corrected electrical signal transmission to the neural center and modulates the corresponding motor activities to warning the IOP increase or decrease.

## Results

### Neuroprosthetic contact lens for point-of-care IOP monitoring and warning

Abnormal intraocular pressure is a major cause of vision loss and eye pathology, which is caused by eye disease, fatigue, and intense physical activity[4]. Because high IOP will compress the optic nerve, resulting in poor vision or even blindness, while persistent low IOP will lead to atrophy of the eyeball and choroidal detachment[21]. Differences in body function make it difficult for groups with poor neural sensitivity to perceive IOP abnormalities, in addition to the slow change in IOP caused by some chronic eye diseases, making timely detection and treatment difficult[5,22]. To sense IOP changes in real time, neuroprosthetic contact lens is designed to modulate neural oscillations between the eye and the brain to evoke a graded nociceptive experience, consisting of a $Ti_3C_2T_x$-SCL for monitoring IOP and an integrated circle for transmitting and encoding information (Fig. 1a, right). Neuroprosthetic contact lens extract information on IOP variations from $Ti_3C_2T_x$-SCL and transmit the feedback via stimulation to the somatosensory cortex, which then processes the information and transmits the resulting commands to the motor cortex thereby generating a corresponding feedback movement. Figure 1b depicts the information transfers from IOP signal generation, to nerve-perception, and to motor activities, forming a neuromorphic artificial loop based on a neuroprosthetic contact lens in the form of a functional component block diagram. In order to accommodate accurate measurements over a wide temperature range, the $Ti_3C_2T_x$-SCL simultaneously acquires both IOP and temperature bimodal signals, which are then encoded and graded by a conditioning circuit, and ultimately the conditioning circuit converts analog signals with different current amplitudes into pulse signals to trigger body feedback.

The elevation of IOP is majorly caused by excessive or impaired outflow of aqueous humor[3] (Fig. 1a, left), which in turn leads to changes in the radius of corneal curvature. In order to provide more accurate IOP sensory feedback, a hemispherical device ($Ti_3C_2T_x$-SCL) based on a Wheatstone bridge structure has been designed that can deform conformally with the cornea to facilitate monitoring of small or slow IOP fluctuations (Supplementary Fig. 1), whilst the ultra-thinness, softness and biocompatibility of the $Ti_3C_2T_x$-SCL allow it to be worn for

extended periods of time for IOP monitoring. Since fluctuations in IOP are closely related to changes in ambient temperature[23], temperature compensation was incorporated into the sensor to provide realistic monitoring of eye conditions.

In addition, a shape-adaptive wet transfer strategy was proposed (Detailed experimental procedures are described in the Materials section, Supplementary Fig. 2). The electrodes are patterned using a maskless laser direct writing technique, which avoids the defects of the mask plate that exerts compressive stress on the electrodes and cause micro-crack expansion within the material, leading to a decrease in conductivity, while skeletonizing the electrode structure overcomes the mechanical limitations of curved surfaces on a two-dimensional substrate. The electrodes were then stripped using a water-soluble film and placed right above the model of a corneal contact lens cured with PDMS, which was repeatedly rinsed to dissolve the film and enhance interfacial adhesion between the electrodes and the PDMS. Subsequently, the transferred electrode was attached onto the hemispherical substrate due to the stronger bonding energy between the substrate and the electrode as the water-soluble film gradually dissolves, whereby this shape-adaptive wet transfer strategy avoids electrode deformation or disconnection caused by the deformation of the 2D substrate in the conventional hot-pressing method. Figure 1c displays an all-MXene-based neuroprosthetic contact lens prepared by this transfer method, with the left photo giving a snapshot of wearing this $Ti_3C_2T_x$-NCL and indicating its excellent biocompatibility while the right image presenting the aesthetics and excellent transparency of the device.

### Material engineering and structural design

Figure 2a illustrates the structure of the $Ti_3C_2T_x$-SCL, which contains two PDMS encapsulation layers and a bimodal sensing electrode layer with a thin thickness. The thickness is approximately 132 μm (Supplementary Fig. 3) with an average mass of 0.031 g (Supplementary Fig. 4 and Supplementary Table 1), which meets the requirements of commercial lenses (<200 μm and -0.037 g). In the center of $Ti_3C_2T_x$-SCL, it has an excellent transparency (94.7%, Fig. 2a III and Supplementary Fig. 5) and an acceptable haze (3.17%, Supplementary Fig. 6). The sensitive electrode layer with an overall thickness of only 57.4 nm is prepared by $Ti_3C_2T_x$-MXene (Fig. 2a II), which provides excellent conductivity while retaining good transparency. MXene was prepared using the mixed acid method with a large lateral size of ~2 μm (Fig. 2a I), excellent conductivity, transparency, biocompatibility, and flexibility, making it an outstanding material for the fabrication of $Ti_3C_2T_x$-SCL. Supplementary Fig. 7 illustrates the X-ray diffraction pattern of the MXene, and the disappearance of the high-angle diffraction peaks and the redshift of (002) peak in the monolayer $Ti_3C_2T_x$-MXene indicates the two-dimensional structure, which provides the flexibility to the electrodes[24]. In addition, owing to the abundance of functional groups on the surface of MXene (Supplementary Fig. 8–11), it possesses excellent water dispersibility and electronegativity, which will induce orderly stacking of nanosheets during the spraying process[25,26]. Meanwhile, in the process of electrode stretching, the nanosheets are prone to relative slippage due to the relatively weaker interlayer-induced van der Waals forces than the chemical bonding, which consequently leads to the change of the electrode resistance[27]. Figure 2e shows the relationship between thickness and resistance at 1% stretching, where the mechanical sensitivity first increased and then decreased as the thickness increased, which came from the combined effect of sliding displacement and interlayer resistance[28,29]. Two-dimensional materials with fewer layers exhibit surface wrinkles and the interlayer friction shows a clear dependence on the number of layers, for which increasing the electrode thickness will suppress the wrinkling effect, resulting in more slip between adjacent sheets due to weaker interlayer friction for the same applied stress and thus achieving more pronounced sensitivity; however, excessive

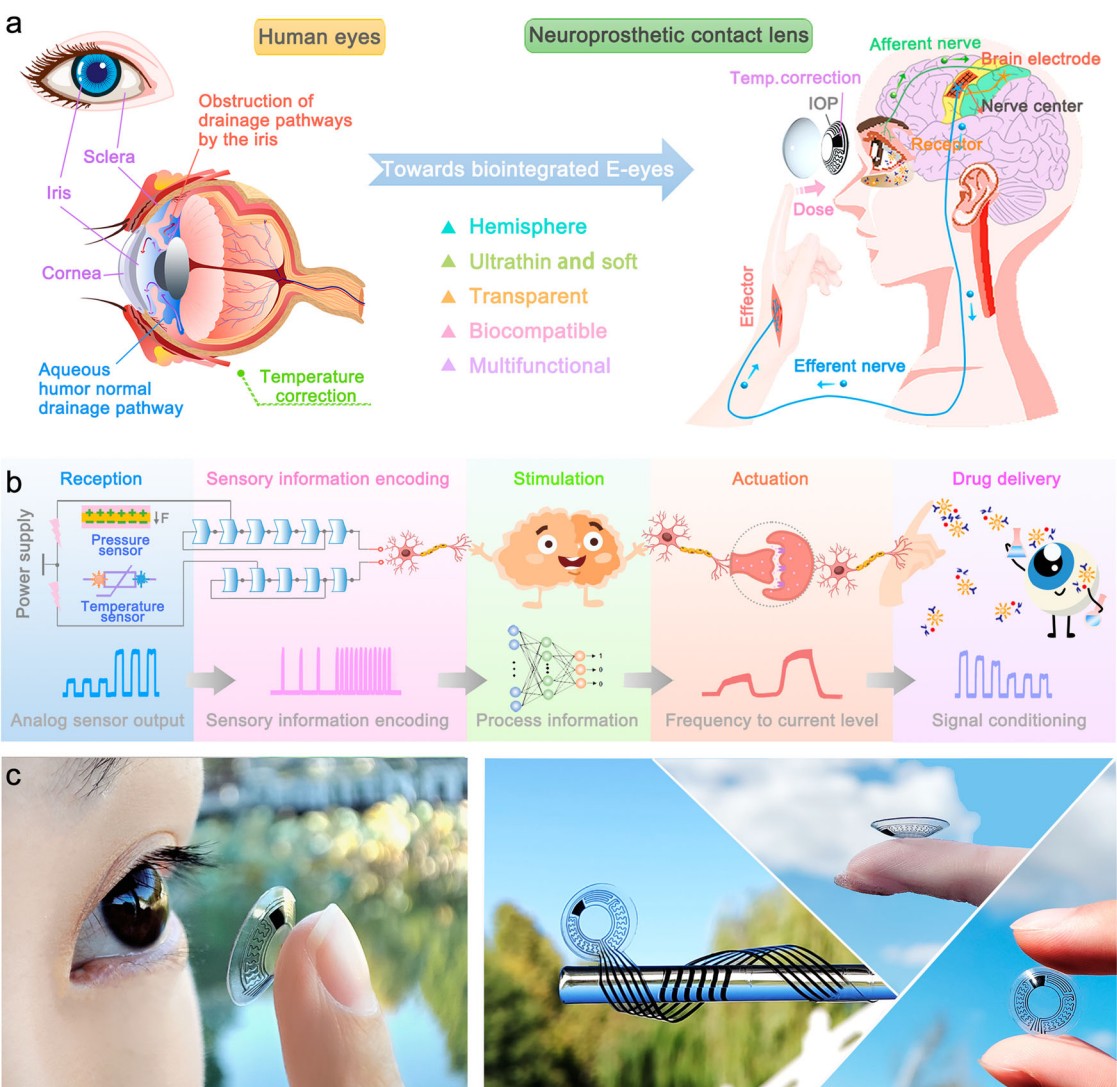

**Fig. 1 | Neuroprosthetic contact lens and the digital photographs of the Ti₃C₂Tₓ soft contact lens. a** Schematic diagram of the human eye and eyeball structure and the structures with the most pronounced deformation of the eyeball at elevated IOP are labeled: cornea, iris, and sclera. The top half shows a schematic of an iris blocking a drainage channel, leading to an increase in intraocular pressure. In contrast, aqueous humor flow in the atrium at normal IOP is presented in the lower half of the diagram. On the right is a neuromorphic artificial circuit based on the neuroprosthetic contact lens. **b** Block diagram of IOP monitoring feedback over a wide temperature range. **c** Photos of Ti₃C₂Tₓ-SCL with the dual function of detecting pressure and temperature.

disordered stacking will limit the slip between nanosheets, resulting in a decrease in sensitivity[30]. Therefore, in order to promote the sensitivity of the IOP sensor, the initial resistance of the electrodes was controlled to be around 300 kΩ.

To provide the sensitivity of IOP, a pressure sensor based on the Wheatstone bridge circuit and stress concentration was designed. The simulation results displayed in Fig. 2c and Supplementary Fig. 12 illustrated that as the IOP increased, the deformation occurred mainly at the corneal-scleral junction, which is primarily caused by the difference in Young's modulus. As IOP increases, the corneal deformation will get greater, which can be detected by the integrated IOP sensor through the conformal deformation of the Ti₃C₂Tₓ-SCL and the cornea (Supplementary Equation 1-10). Moreover, the corneal deformation during the increase of IOP is uneven, with a maximum of 5 mm from the center of the cornea and gradual decrease in the radial direction (Fig. 2b), while the deformation in the ring direction is larger than that in the radial direction (Fig. 2d). Based on the above simulation results, the strain arm and reference arm of the Wheatstone bridge are designed in the circumferential and radial positions, respectively,

which also skillfully avoid obstructing the field of view range (Fig. 2f). To further increase the sensitivity of the IOP sensor, the strain gauge was designed as rectilinear to amplify the resistance change during deformation, while the reference gauge was designed as a serpentine to attenuate the change in resistance during deformation. The simulation results and test results further verified that the resistance change of the rectilinear electrode is much larger than that of the serpentine electrode under the same tensile strain condition (Fig. 2g, h and Supplementary Fig. 13, 14). Combined with the above material selection and structural optimization design, the MXene-based IOP sensor has a wide detection range and high sensitivity, which is the highest sensitivity among the reported IOP sensors (Fig. 2i, j and Supplementary Table 2)[8,9,31–39]. Metal electrodes such as platinum-titanium and copper have fewer interatomic contact sites that can be separated under lower strains, resulting in a smaller pressure test range, while the smaller specific surface area of bulk phase materials results in lower sensitivity of the prepared sensors. Comparatively, two-dimensional materials like graphene and MXene have a high specific surface area and interlayer van der Waals forces, which can cause slip and result in good

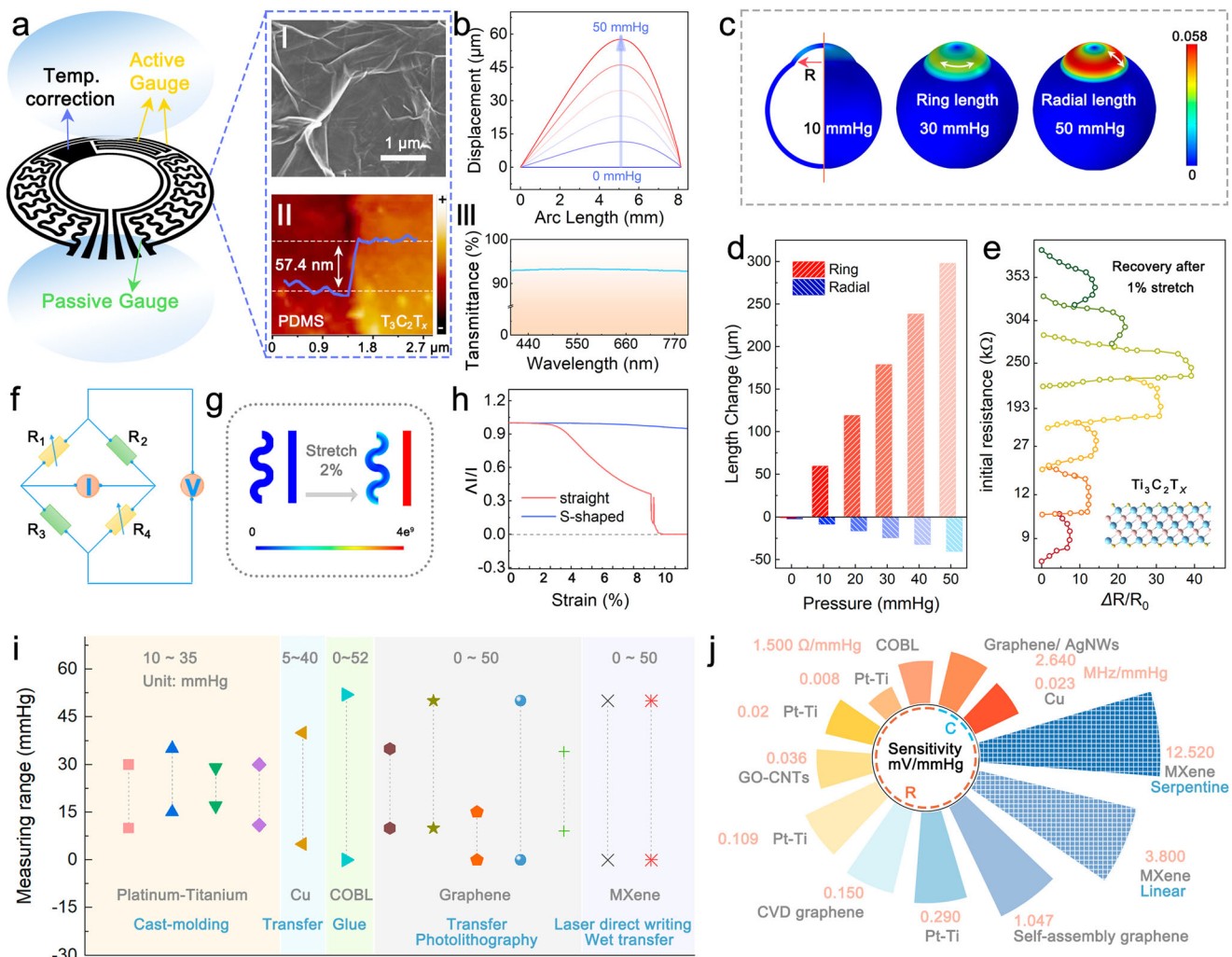

**Fig. 2 | Construction and structural design of the Ti₃C₂Tₓ-SCL for IOP monitoring. a** The structure of the Ti₃C₂Tₓ-SCL. Panel 1 shows the SEM image of the Ti₃C₂Tₓ MXene electrode functional material. Panel II displays the thickness of the functional electrode layer as measured by AFM. Panel III highlights the high transparency in the Ti₃C₂Tₓ-SCL viewing area. **b** Relationship between corneal basal arc at distance from axial deformation (R) and IOP. **c** Surface stress distribution of the eyeball under different IOPs (10, 30, and 50 mmHg), with the cornea in the upper half and the sclera in the lower half. The circumferential and radial directions of the cornea are clearly labeled. **d** Relationship of the corneal base arc deformation in the ring and radial directions to IOP. **e** Effect of electrode thickness on sensitivity at 1% strain recovery. The inset presents the structure of the Ti₃C₂Tₓ MXene, where blue indicates Ti atoms, pink indicates C atoms and yellow indicates surface functional groups. **f** Schematic diagram of a Wheatstone bridge circuit, where $R_1$ and $R_4$ represent the active gauges as well as $R_2$ and $R_3$ represent the passive gauges. **g** Finite element analysis of rectilinear and serpentine electrodes under the original state and 2% strain. **h** Current variation versus tensile strain for serpentine and rectilinear electrodes. Comparison of previously reported corneal contact lens IOP sensors with our work in terms of measurement range (**i**) and sensitivity **j**, where Pt-Ti, GO-CNTs, and COBL are defined as platinum-titanium, reduced graphene oxide and carbon nanotubes and conductive all-organic bilayer film respectively.

stretchability and high detection sensitivity. The Ti₃C₂Tₓ MXene combined with the excellent structural design enabled the IOP sensors with outstanding performance, which is promising for potential applications in fields such as neuroprosthetics.

## Performance evaluation of Ti₃C₂Tₓ-SCL

A simulated intraocular pressure testing platform containing a pressure adjustment system, a pressure testing system, and a pressure feedback system was constructed to evaluate the accuracy and speed of sensory feedback provided by the neuroprosthetic contact lens (Fig. 3a and Supplementary Fig. 15, 16). In order to investigate the detection accuracy of the sensor, the pressure was applied to the bionic eyeball at a rate of 1 r s⁻¹ (1x) up to 50 mmHg and then reduced to the initial state in the same way, during which the pressure was stopped for 10 s for each change of 6.25 mmHg. Figure 3b and Supplementary Fig. 17 depict the output voltage change (ΔV) curve is

stepped and each step corresponds to the variation of the pressure curve, and further statistical analyses (Supplementary Fig. 18–20 and Supplementary Table 3) demonstrate that the Ti₃C₂Tₓ-SCL has a minimum detection limit and an effective resolution of both 0.05 mmHg with a measurement accuracy of 1.075% at 0.2 mmHg, which confirms that small IOP changes can be accurately recognized. IOP will fluctuate to varying degrees owing to a number of factors, such as postural change, day and night alternation, emotional highs, and eye diseases. It is important that Ti₃C₂Tₓ-SCL was able to respond stably and significantly multiple times to pressure changes of different magnitudes based on the results of Fig. 3d, with the change in output voltage (ΔV) showing a good linear relationship with pressure (linear regression coefficient $R^2$ = 0.998), and achieving a high sensitivity of 12.52 mV mmHg⁻¹ (Fig. 3c) and a small error of 0.029 mV (Supplementary Fig. 21 and Supplementary Table 4) during pressure increase. To further examine the responsiveness of the sensor, cycling tests

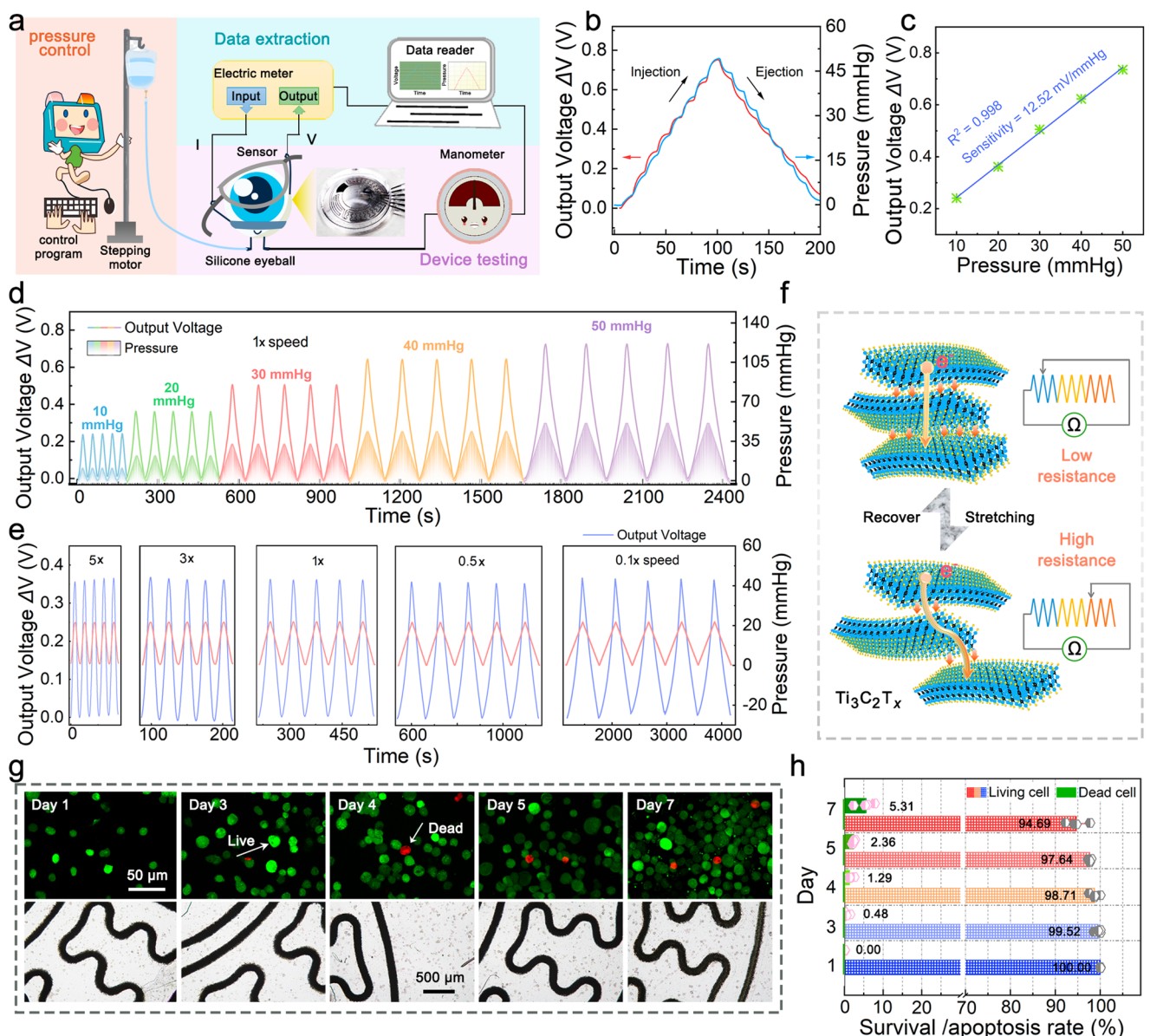

**Fig. 3 | Performance analysis and biocompatibility measurement of the $Ti_3C_2T_x$-SCL. a** The cartoon schematic of a simulated IOP testing platform with three modules: pressure control, device testing and data extraction. The inset is a photograph of the bionic eyeball. **b** Static response of the $Ti_3C_2T_x$-SCL at 1x speed with pressure change in steps of 6.25 mmHg. **c** The fitted relationship between the output voltage variation and the pressure, where each point is averaged over 5 cycles at the corresponding pressure. **d** Output voltage variation curves at different pressure amplitudes (0–10, 0–20, 0–30, 0–40, 0–50 mmHg) at 1x speed. **e** Output

voltage variation curves within the range of 0-20 mmHg at 5x, 3x 1x, 0.5x and 0.1x speed. **f** Microscopic schematic of $Ti_3C_2T_x$ MXene nanosheets in the initial state and under tensile stress as well as changes in the electron transport channels where blue indicates Ti atoms, black indicates C atoms and yellow indicates surface functional groups. The modeling of the sensing mechanism corresponds to a sliding varistor. **g** Fluorescent and optical photographs of cytotoxicity tests on days 1, 3, 4, 5, and 7, as well as (**h**) survival and apoptosis rate statistics.

were performed at different rates (0.1x, 0.5x, 1x, 3x, 5x) over the range of IOP changes from 0 to 20 mmHg. Figure 3e shows that the output voltage change is synchronized with the pressure change without significant delay, and the amplitude of the voltage change ($\Delta V$) is relatively stable at different rates of pressure change, indicating that the sensor can respond accurately to IOP fluctuations with different rates. In addition, the device was cycled at 1x speed for 1 h within the range of 0–21 mmHg and there was no significant performance degradation (~3%), suggesting that the device may be suitable for long-term monitoring (Supplementary Fig. 22). The excellent performance of the sensor is based on two main reasons: (1) The slip phenomenon based on the action of weak van der Waals forces occurring in $Ti_3C_2T_x$ MXene nanosheets under stress reduces electron transport channels

and increases electrode resistance. Meanwhile, the friction that determines the slip degree induces a high dipole potential on the interlayer potential energy surface under applied loads, thus inhibiting the electron transport behavior between the layers[40,41], as reflected in Fig. 3f. The clever bridge design and stress concentration strategy amplifies corneal strain and improves sensitivity (Supplementary Fig. 23).

To evaluate the biocompatibility of the $Ti_3C_2T_x$-SCL, a 7-day in vitro cytotoxicity assay was performed using human umbilical vein endothelial cells (HUVECs), which are similar to retinal neovascularization endothelial cells and capable of unlimited passaging. Figure 3g, h shows that the cells were evenly distributed on the $Ti_3C_2T_x$-SCL and their number was increasing with a proliferation rate of 309%

on day 7, accompanied by a significantly larger number of live cells than dead cells, with a survival rate of over 94% within one week, meeting the requirements for cell culture (Supplementary Fig. 24–26 and Supplementary Table 5, 6). The rabbit was also worn for 7 consecutive days and performed slit lamp examinations and white light photographs of the eyes, which depicts in Supplementary Fig. 27–29 and Supplementary Video 1 that the $Ti_3C_2T_x$-SCL is not significantly harmful to eyes and can be worn for extended periods. In summary, the $Ti_3C_2T_x$-SCL can provide comprehensive and accurate IOP sensory feedback due to its excellent performance and good biocompatibility.

**Temperature correction over a wide range from $-20\,°C$ to $40\,°C$**

$Ti_3C_2T_x$-SCL must possess high sensitivity and a wide detection range, as well as the ability to perform stable measurements at any ambient temperature (Fig. 4a). All the resistors in $Ti_3C_2T_x$-SCL are made of $Ti_3C_2T_x$ MXene, but the difference of the nanosheet slip distance in the different resistors caused by the inhomogeneous corneal deformation during the IOP change leads to different temperature coefficients from each other, which in turn leads to errors in the measurement results under variable temperature conditions (Supplementary Eq. (11–16))[42]. $Ti_3C_2T_x$ MXene nanosheets exhibit a high friction factor of $3.9 \pm 0.1$, which is attributed to interlayer adhesive hydrogen bonding and the rigidity properties of the metal layer, and undergo friction-induced in-

plane lattice distortions including lattice structure torsion, increased atomic spacing, or lattice site dislocations when slip occurs in the presence of tangential forces[43]. For 2D van der Waals semiconductors, the lattice distortion-induced interlayer electronic coupling induces a change in the density of states at the Fermi energy level and leads to a change in the material bandgap, which in turn leads to different temperature coefficients of the $Ti_3C_2T_x$ MXene resistance under different stresses[44,45].

Accordingly, a $Ti_3C_2T_x$ MXene-based temperature sensor was fabricated with high sensitivity (TCR = −0.996%) and good linearity ($R^2 = 0.99877$) over the temperature range of −20 to 20 °C for real-time monitoring of ambient temperature as displayed in Fig. 4c. $Ti_3C_2T_x$ MXene with abundant surface groups (−O, −OH and −F) exhibits narrow band gap semiconductor properties and contact resistance between the nanosheets, and an elevated temperature promotes the electron tunneling effect, which improves the electron transport efficiency and increases the carrier concentration (Fig. 4b)[46]. The cycling test at −5 °C and 5 °C in Fig. 4e illustrates the accuracy and durability of the sensor for variable-temperature use, with a response variation of only 0.367% at −5 °C and 0.797% at 5 °C, respectively. The dynamic response of the temperature sensor was also measured by slowly increasing the temperature at a rate of 1 °C min⁻¹, as shown in Supplementary Fig. 30, it can be seen that the dynamic current response is

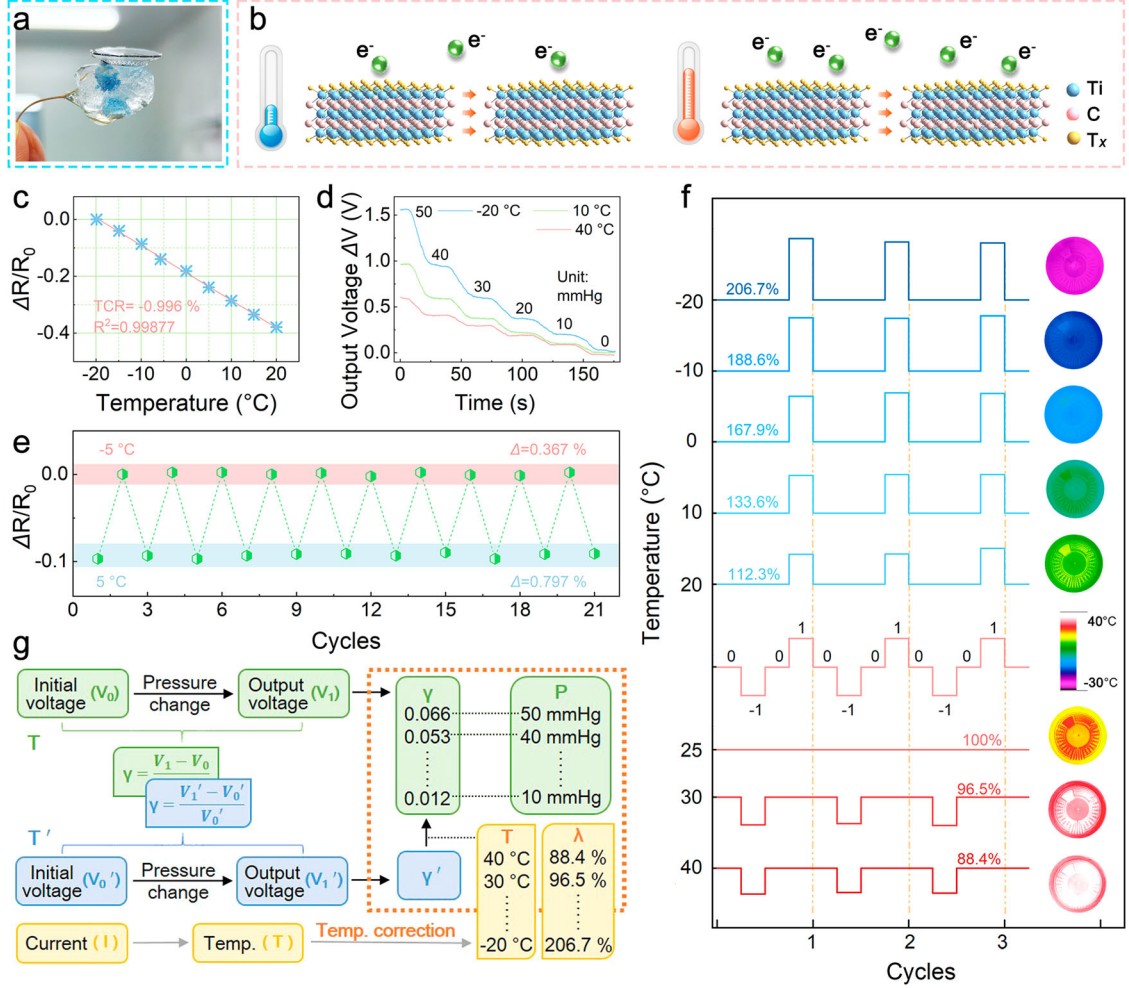

**Fig. 4 | Performance characterization of $Ti_3C_2T_x$ MXene temperature sensor and temperature correction of $Ti_3C_2T_x$-SCL. a** Photograph of the $Ti_3C_2T_x$-SCL on an ice flower, demonstrating its robustness for use at low temperatures. **b** Temperature sensing mechanism of the $Ti_3C_2T_x$ MXene-based electrode. **c** Static test and fitting results of temperature sensors. **d** IOP gradient test at different temperatures. **e** Cycling test at −5 °C and 5 °C. **f** The sensitivity change rate of $Ti_3C_2T_x$-SCL versus temperature over the IOP range of 0–50 mmHg. The insert displays thermal images of the $Ti_3C_2T_x$-SCL at various temperatures (Supplementary Video 2). **g** Flow chart for temperature compensation in an IOP monitoring system.

consistent with the increase of the temperature, indicating that there is almost no time lag in the detection of the temperature signal. As a result, the $Ti_3C_2T_x$-SCL simultaneously senses both temperature and pressure bimodal signals and outputs accurate IOP results over a wide temperature range. Interestingly, Fig. 4d demonstrates that the $Ti_3C_2T_x$-SCL can clearly distinguish IOP changes at different temperatures, but the output voltage change for the same IOP tested at low temperatures is significantly higher than the results tested in a high-temperature environment. Cycling tests were performed on the $Ti_3C_2T_x$-SCL in the range of 0-50 mmHg at different temperatures as presented in Fig. 4f, the sensitivities at −20 °C and 40 °C are 206.7% and 88.4%, respectively, of those at ambient temperature (25 °C), and there has a good linear relationship between the sensitivities and temperatures (Supplementary Fig. 31). Therefore, the temperature compensation method of the $Ti_3C_2T_x$-SCL is specified in Fig. 4g on the basis of the above-mentioned test results. The output voltage difference of the $Ti_3C_2T_x$-SCL between the pressure and zero-pressure states corresponds directly to the value of IOP in a room-temperature environment. The output voltage difference in a variable temperature environment under the same IOP fluctuation differs from the output voltage difference at room temperature. At this point, the external circuit adjusts the output voltage based on the ambient temperature measured by the temperature sensor to obtain an accurate IOP value.

### Demonstration of the neuromorphic sensorimotor loop

The transmission of nerve impulses in the nervous system during IOP abnormalities was investigated in a live rat model (Fig. 5a and Supplementary Fig. 32, 33). There were no changes of potentials in the somatosensory cortex when high IOP was induced by saline injection into rat eyes based on the result of Supplementary Fig. 34, indicating weak neural oscillations between the eye and the brain. The neuroprosthetic contact lens was connected to the brains of rats, and nerve impulses were successfully observed in the somatosensory cortex when IOP was abnormal, establishing limb sensation related to IOP. Finally, the control signals were processed by the cerebral cortex and transmitted to the motor cortex which in turn governs somatic movements, completing the artificial sensorimotor circuit (Fig. 5a III, Supplementary Figs. 35–37). The whole process was provided in Supplementary Video 3, which obviously displays the rat leg twitching under the control of the somatosensory cortex when IOP levels were abnormal. However, the graded potential changes in the somatosensory cortex led to a small leg twitching angle, the angular changes between different IOP levels could not be well observed at a small twitching angle, therefore we added a supplemented test by direct stimulation of the sciatic nerve to amplify the effect of leg movements. In this system of Fig. 5b and Supplementary Fig. 38, the $Ti_3C_2T_x$-SCL simultaneously senses both temperature and pressure bimodal signals and collects them via an analog-to-digital converter (ADC) integrated within the microcontroller unit (MCU). The MCU decodes the status of the intraocular pressure by processing pressure signals based on the temperature, which acts as a neural oscillation between the eye and the brain. Subsequently, the DAC module built into the MUC regulates the output current of the voltage-controlled current source based on the IOP level and then switches the current through analog switches to produce a pulsed current stimulation signal that triggers the corresponding action or bodily response, which is similar to the signal processing in the central nervous system and the sensing process of the evoked neurotransmitters in the synapses. Meanwhile, the MUC can transmit data to the mobile APP via Bluetooth to realize real-time IOP monitoring. The experiment simulated different eye conditions by controlling the pressure within the bionic eye, including low intraocular pressure (0-9 mmHg), healthy intraocular pressure status (10–21 mmHg), and high intraocular pressure (22–30, 31–40, and 41–50 mmHg). The MUC classifies the signals detected by the $Ti_3C_2T_x$-SCL and stimulates the sciatic nerve depending on the IOP level to

contract the gastrocnemius muscle at a frequency equal to 1 Hz (Supplementary Fig. 39, 40). As shown in Fig. 5c, Supplementary Fig. 41 and Supplementary Video 4, there is no stress response in the leg of the rat under normal IOP, but the leg exhibits a slight angle of flexion, approximately 22° when the intraocular pressure is below normal. In contrast, when the eye is under high IOP conditions, the leg exhibits a greater angle of flexion, which increases with increasing IOP values, reaching 44° at 22-30 mmHg, 62° at 31–40 mmHg, and 76° at 41–50 mmHg. The signal intensities corresponding to different IOP levels acquired on the gastrocnemius muscle demonstrate that the neuroprosthetic contact lens based on the $Ti_3C_2T_x$ MXene can provide real-time and graded sensory feedback (Fig. 5d).

## Discussion

We reported a neuroprosthetic contact lens enabled sensorimotor system to realize the point-of-care IOP monitoring and real-time display. The neuroprosthetic contact lens was based on 2D $Ti_3C_2T_x$ MXene with specially designed serpentine architecture, which exhibited a high sensitivity of 12.52 mV mmHg$^{-1}$ and excellent stability. It could convert the IOP information to an electric signal and after corrected with a $Ti_3C_2T_x$ MXene temperature sensor, the corrected electrical signal was transferred to the neural center via a circuit board, which finally modulated the corresponding motor activities to warn the IOP variation, thus forming a closed-loop of IOP signal generation−nerve-perception−motor activities. To verify the biocompatibility and potential bio-application of the neuroprosthetic contact lens, it was inserted into the eyes of a rabbit in vivo and connected with a live rat in vitro, respectively. The results revealed that the designed neuroprosthetic contact lens has excellent biocompatibility and can successfully monitor the variation of IOP signals. In future research, exploring more rational ways of signal acquisition and energy supply, such as integrating both processing chips and sensors into contact lenses, will greatly enhance the comfort and utility of IOP monitoring devices and provide solutions for further development into neuroprosthetic devices.

## Methods

### Materials and synthesis

$Ti_3AlC_2$ MAX was provided by Carbon-Ukraine. Hydrofluoric acid (HF) and hydrochloric acid (HCl) were purchased from Tianjin Fuyu Fine Chemical Co., Ltd. N,N-Dimethylformamide (DMF), Lithium chloride (LiCl) and polyethylene oxide (PEO; MW = 100,000) were supplied by Innochem. The P(VDF-TrFE) (VDF: TrFE = 80%: 20%) was procured from France PIEZOTECH. All reagents were analytically pure.

$Ti_3AlC_2$ MAX was etched with mixed acids (HF and HCl) with magnetic stirring, and multilayer $Ti_3C_2T_x$ MXene was obtained by hand-holding cleaning. The multilayer $Ti_3C_2T_x$ MXene was then intercalated by LiCl and cleaned by centrifugation to obtain monolayer 2D $Ti_3C_2T_x$ MXene nanosheets[47].

### Device fabrication

The solutions were pre-prepared before fabricating. 20 mg 2D $Ti_3C_2T_x$ MXene powder and 2 g P(VDF-TrFE) were successively dissolved in 13.3 mL DMF and magnetically stirred for 8 h to obtain P(VDF-TrFE)@ $Ti_3C_2T_x$ solution. 0.5 g of PEO was added to 10 g DI-water and then under magnetic stirring at 50 °C for 3 h. Polydimethylsiloxane (PDMS) containing 90 wt.% main agent and 10 wt.% curing agent was mixed well and placed in refrigerator for later use.

Initially, a layer of P(VDF-TrFE)@ $Ti_3C_2T_x$ solution was spin-coated on a cleaned slide at 1500 rpm for 20 s. The slide was dried on a hot plate at 35 °C for 10 min, naturally cooled to room temperature and set aside. 200 μL 2D $Ti_3C_2T_x$ MXene at a concentration of 1 mg/mL was sprayed onto the P(VDF-TrFE)@ $Ti_3C_2T_x$ film. The distance between the spray gun and the film was 20 cm, and the whole process was carried out on a hot plate at 60 °C. Subsequently, the film was

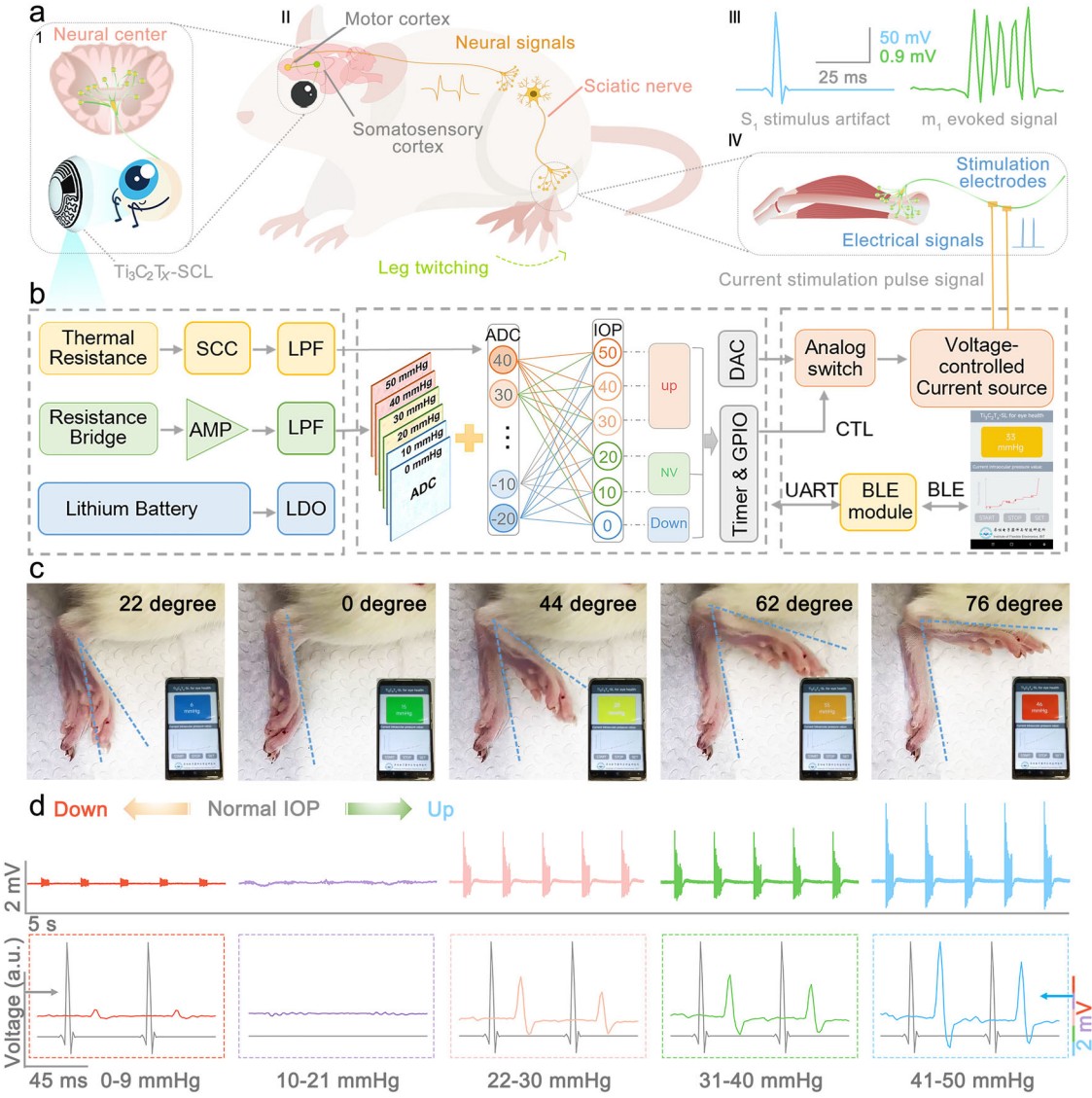

**Fig. 5 | Neuromorphic CL for IOP sensing-motion feedback loop. a** Schematic diagram of the neural loop for IOP sensing and feedback. Panel 1 depicts a neuro-prosthetic contact lens that provides sensory feedback to the brain. Panel II is a schematic diagram of the neural reflex circuit for sensing IOP. Panel III records the stimulation signals to the somatosensory cortex and the corresponding potential signals collected in the motor cortex during abnormal intraocular pressure in rats. Panel IV shows a schematic diagram of sciatic nerve stimulation to induce contraction of the gastrocnemius muscle. **b** System block diagram of the pressure and temperature signals processing step including signal acquisition, processing, control, communication and display. **c** Photographs of leg twitch responses in a live rat

model and real-time IOP display stimulated at various simulated IOP levels (From left to right, the images show low IOP in the range of 0-9 mmHg, normal IOP in the range of 10–21 mmHg, and high IOP in the ranges of 22–30, 31–40, and 41–50 mmHg, respectively). **d** Signals acquired in the gastrocnemius muscle while stimulated at various IOP levels in a living rat model (The red, purple, pink, green, and blue curves correspond sequentially to data collected in the gastrocnemius muscle when IOP is in the range of 0–9 mmHg, 10–21 mmHg, 22–30 mmHg, 31–40 mmHg, and 41–50 mmHg. The colored curves in the dashed box in the second column are locally magnified graphs of the signals in the first column. The gray curve shows the stimulation signal of the NCL).

patterned by laser direct writing technique and the excess sensor pattern was removed. Next, PEO was spin-coated on the slide with the sensor at 500 rpm for 10 s and dried at 60 °C for 30 min on a hot plate. Following this, the PEO was peeled off the slide and the sensor was transferred to the PEO film. At the same time, PDMS was drop-coated onto the contact lens mold and dried overnight at room temperature. In a particularly important step, the PEO film with the sensor was placed vertically aligned onto the contact lens mold coated with PDMS. The whole device was rinsed under a water tap, and the PEO gradually dissolved in the water and the sensor was attached to the PDMS. Encapsulated again with PDMS and decapsulated to obtain soft and transparent $Ti_3C_2T_x$-SCL (Fig. S1).

## Cell culture and biocompatibility studies

HUVEC cells were first revived by heating in a water bath and then cultured in a complete medium containing 10% fetal bovine serum. When the cell density reached 80%, the cells were passaged at a ratio of 1:3. After that, the HUVEC cells in the logarithmic growth phase with a good growth status were inoculated into the 6-well plates of the cell culture, and the glasses were placed on the bottom of the 6-well plates, respectively. Each group received 500 μl of cell suspension in the middle of the material and cultured in the incubator for the specified time. When the cell culture was complete, 50 μL of CCK8 was added to the cell suspension in each well of the material, and the absorbance was measured using an enzyme-labeled OD 450.

## Surgical procedure in rats

All experiments on rats mentioned in this article were repeated with three rats as necessary and conducted in Beijing Jinglai Huako Biotechnology Co. with the permission of Beijing Administration Office of Laboratory Animal (approval number: 202340191 and JLSW-20231220-4). The experiment was performed on ten-week-old male SD rats weighing approximately 300 g with normal vital signs. Prior to surgery, the rats were anesthetized by injection of 3% pelltobarbitalum natricum. After the rats were deeply anesthetized as determined by observation of whiskers and pinching of toes, the head was shaved with a surgical blade and a midline skin incision was made in the skull, followed by perforation of the skull with a dental drill to expose the somatosensory and motor cortex. For sciatic nerve surgery, a 2 cm skin incision was made approximately 1.5 mm anterior to the femur in the rat hind leg, and then the sciatic nerve was exposed by slowly and repeatedly separating the muscles near the femur with forceps. They were executed at the end of the experiment using the cervical dislocation method.

## Reporting summary

Further information on research design is available in the Nature Portfolio Reporting Summary linked to this article.

## Data availability

The data supporting the plots in this paper are available from the corresponding author upon request. Source data are provided with this paper.

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

## Acknowledgements

This work was supported by the Beijing Natural Science Foundation (L223006), the Beijing Institute of Technology Research Found Program for Young Scholars and the BIT Research and Innovation Promoting Project (2023YCXZ010).

## Author contributions

W.L. and L.L. conceived the concept and designed the experiments. W.L. prepared the materials and the fabricated devices. W.L., L.L., and G.S. characterized the device performance and analyzed the experimental data. W.L., L.L., and G.S. wrote the paper. Z.J.D. and Z.Y.D. revised the paper. All authors reviewed and revised the paper.

## Competing interests

The authors declare no competing interests.
