## [Peer Review File · Nature Communications]

Neuroprosthetic contact lens enabled sensorimotor system for point-of-care monitoring and feedback of intraocular pressureREVIEWER COMMENTS

Reviewer #1 (Remarks to the Author):

The manuscript written by Weijia Liu et al. reported a neuroprosthetic contact lens for point-of-care monitoring and feedback of intraocular pressure, which consists of a smart contact lens with Ti3C2Tx Wheatstone bridge structured IOP strain sensor, a Ti3C2Tx temperature sensor, and an IOP point-of-care monitoring/display system. The high sensitivity of the neuroprosthetic contact lens is ~ 12.52 mV mmHg⁻¹, ensuring the IOP monitoring and warning. The wearability and biocompatibility of the neuroprosthetic contact lens have been demonstrated in vivo experiments on rabbit eyes. In addition, the biological sensorimotor loop was successfully mimicked by the experiments on a living rat in vitro. This work is interesting and has significance for the treatment of eye disease and visual restoration. Thus, the manuscript can be published after addressing the following issues via minor revision.

(1) The authors demonstrated the high transparency in the Ti3C2Tx-SCL viewing area. However, haze is also an important factor for contact lenses. Thus, the authors are advised to measure the haze of the prepared neuroprosthetic contact lens.

(2) The center thickness of Ti3C2Tx-SCL was measured with a film thickness gauge, which was between 130 ~ 140 μm , meeting the requirements for commercial lenses (< 173 μm). The authors are advised to measure the weight of the Ti3C2Tx-SCL and compare it with the commercial lenses.

(3) The Ti3C2Tx-SCL with a high sensitivity of 12.52 mVHg⁻¹ can respond stably and significantly multiple times to pressure changes of different magnitudes. The authors are advised to give the smallest detection limit of Ti3C2Tx-SCL.

(4) In this paper, the authors prepared a neuroprosthetic contact lens for monitoring and feedback of intraocular pressure and further demonstrated the neuromorphic sensorimotor loop. The advantages of the neuroprosthetic contact lens and neuromorphic sensorimotor loop have been elaborated. However, the disadvantages of these devices and systems and corresponding improved methods should also be discussed.

(5) In recent years, some studies about retinomorphic neurons and artificial reflex arcs based on neuromorphic devices for artificial vision or visual restoration have been reported. These works may have important applications in visual neuroprosthetics, thus the authors are advised to make a brief introduction to these artificial neurons and artificial reflex arcs. (Ref: Nature Communications 2023, 14, 7181; Nature Communications 2022, 13, 6760; Matter 2022, 5, 1578-1589; Materials Horizons 2023, 10, 5753-5762.)

Reviewer #2 (Remarks to the Author):

This work presented a neuroprosthetic contact lens enabled sensorimotor system, which consisted of a Ti3C2Tx IOP strain sensor, a Ti3C2Tx temperature sensor and an IOP monitoring/display system, to realize the point-of-care IOP monitoring and real-time display. With the IOP strain sensor, the system could convert the IOP information into an electrical signal, which was then corrected by the temperature sensor to obtain and display an accurate IOP value. Further, according to the level of IOP, the system could generate corresponding electrical signals to stimulate the neural center, and eventually formed a closed-loop of IOP signal generation-nerve-perception-motor activity. Finally, the authors demonstrated the good biocompatibility of the designed neuroprosthetic contact lenses through rabbit eye experiments, and verified the feasibility of the closed-loop IOP neurosensory activity through rat experiments. Based on the unique system design and the compelling IOP monitoring results, this manuscript is acceptable after major revision. The following comments need to be addressed before further consideration.

1) Normally, IOP would fluctuate cyclically due to a variety of factors such as day and night changes, emotional changes, and eye diseases. Accurately obtaining the upward and downward trends of IOP is necessary for the health monitoring of patients with eye diseases. From Figure 3b and Supplementary Figure 15a, it can be seen that under the same IOP value, there is a certain degree of difference between the output voltage of the Ti3C2Tx IOP strain sensor in the rising phases and the falling phases. This means that the sensor is unable to accurately obtain IOP values based on a single-point test alone when the rising and falling trends are unknown, which is unacceptable for IOP monitoring in patients with eye diseases.

2) The neuroprosthetic contact lens enabled sensorimotor system, as a point-of-care IOP monitoring and real-time display system, needs to consider the signal transmission and power supply of the Ti3C2Tx-SCL. Considering the wearing comfort and practicality in daily life, there are

many excellent works that use wireless transmission or self-powered forms to solve the signal acquisition and power supply problems of IOP sensors. In this work, how was the issue of signal acquisition and power supply addressed? How does this affect the further development of this system into a wearable IOP monitoring system?

3) This work verified that the detection range (0-50 mmHg) and detection sensitivity (12.52 mV/mmHg-1) of the IOP strain sensor could meet the needs of both the normal population and the ocular disease patient population. However, the accuracy and measurement error of the IOP stress sensor is not statistically analyzed in the manuscript. In addition, what is the effective resolution of IOP strain sensors during the IOP detection? The authors should conduct further investigation on these aspects.

4) During the sensing process of IOP strain sensor and temperature sensor, is there a time lag (i.e., a hysteresis in the sensor's response) in the detection of the IOP signal and the temperature signal? If so, does the time lag differ between IOP detection and temperature detection? And how would this difference affect the accuracy of the final IOP results?

5) How does the neuroprosthetic contact lens enabled sensorimotor system connect with the somatosensory cortex, motor cortex, and sciatic nerve to generate electrical stimulation during in vivo experiments in rats? And how were the corresponding potential signals collected at the different positions? The authors should describe the corresponding parts of the article in more detail.

6) In this work, a neuromorphic sensorimotor loop design was proposed (Figure 5a). However, in the subsequent actual experiment (Fig. 5c), it seems that the authors directly electrically stimulated the sciatic nerve through the circuit to realize the effect of controlling the movement of the leg, without the involvement of the somatosensory cortex and the motor cortex of the brain? Could the authors please provide a reasonable explanation for the discrepancy between the description of the neuromorphic sensorimotor loop and the actual experimental test?

7) Figure 5d was confusing, and it should be clearly labeled and explained what the different colored curves represent. In addition, there is no difference in amplitude and frequency between the electrical stimulation signals (black curves) in various IOP ranges, which was obviously not in accordance with the actual situation. The authors should make a clear revision and explanation of Figure 5d.

8) This manuscript consisted of two main parts: one was the preparation and performance testing of an IOP strain sensor with temperature correction; the other was a neuromorphic sensorimotor loop for IOP monitoring through stimulation of the neural center. In this work, after obtaining IOP information from the IOP strain sensor and estimating whether IOP levels were abnormal, the authors used an external circuit system to electrically stimulate the somatosensory cortex, sciatic nerve, and other parts of the brain for neuromorphic sensorimotor feedback. While the two parts of the work were convincingly presented and achieved good results, the whole work was more like two separate pieces of work cobbled together. The authors should fully explain the intrinsic connection between the two pieces of work to better demonstrate the integrity and consistency of the manuscript.

9) The authors are suggested to improve the English of the manuscript. It is better to be edited by native English speakers.

Responses to the referees' comments:

Responses to Reviewer 1:

The manuscript written by Weijia Liu et al. reported a neuroprosthetic contact lens for point-of-care monitoring and feedback of intraocular pressure, which consists of a smart contact lens with $Ti_3C_2T_x$ Wheatstone bridge structured IOP strain sensor, a $Ti_3C_2T_x$ temperature sensor, and an IOP point-of-care monitoring/display system. The high sensitivity of the neuroprosthetic contact lens is $\sim 12.52 \text{ mV mmHg}^{-1}$, ensuring the IOP monitoring and warning. The wearability and biocompatibility of the neuroprosthetic contact lens have been demonstrated in vivo experiments on rabbit eyes. In addition, the biological sensorimotor loop was successfully mimicked by the experiments on a living rat in vitro. This work is interesting and has significance for the treatment of eye disease and visual restoration. Thus, the manuscript can be published after addressing the following issues via minor revision.

Answer: We thank the reviewer very much for the positive comment on our work as well as the valuable suggestions. Responses according to reviewer's suggestions are provided point-by-point as following

Comment 1: The authors demonstrated the high transparency in the $Ti_3C_2T_x$ -SCL viewing area. However, haze is also an important factor for contact lenses. Thus, the authors are advised to measure the haze of the prepared neuroprosthetic contact lens.

Answer: We greatly appreciate the referee's professional questions. Haze is an important parameter for corneal contact lenses, and an increase in haze will result in a decrease in gloss as well as transparency and especially imaging. Haze is the cloudy or cloudy appearance of the interior or surface of a transparent or semi-transparent material due to light diffusion, expressed as a percentage of the ratio of the diffused luminous flux to the luminous flux through the material. Therefore, the transmittance and haze of the neuroprosthetic contact lens were tested as 94.7% and 3.17% at 550 nm by ultraviolet-visible absorption spectroscopy (UV-3600), respectively.

Supplementary Fig. 5. Transmittance of the $Ti_3C_2T_x$ -SCL.

Supplementary Fig. 6. Haze of the $\text{Ti}_3\text{C}_2\text{T}_x$ -SCL.

Comment 2: The center thickness of $\text{Ti}_3\text{C}_2\text{T}_x$ -SCL was measured with a film thickness gauge, which was between 130 ~ 140 μm , meeting the requirements for commercial lenses (<173 μm). The authors are advised to measure the weight of the $\text{Ti}_3\text{C}_2\text{T}_x$ -SCL and compare it with the commercial lenses.

Answer: We thank the reviewer for raising the good question. 10 random devices were weighed and a mean value of 0.0314 g was calculated, which shows lighter weight than 0.041 g for transparent contact lenses (TCL) and 0.364 g for color contact lenses (CCL) from Bausch & Lomb, 0.0355 g for transparent contact lenses and 0.0392 g for color contact lenses from Cooper Optics, and 0.0339 g for transparent contact lenses and 0.0370 g for color contact lenses from Hydron (Table S1 and Fig. S4). The lighter weight of the $\text{Ti}_3\text{C}_2\text{T}_x$ -SCL meets the requirements of commercial lenses and ensures wearing comfort.

Supplementary Table 1: Weight comparison of $\text{Ti}_3\text{C}_2\text{T}_x$ -SCL with commercial transparent and color contact lenses (Unit: g).

Decive	Cooper Optics		Bausch & Lomb		Hydron		$\text{Ti}_3\text{C}_2\text{T}_x$ -SCL
	CCL	TCL	CCL	TCL	CCL	TCL	
1	0.0394	0.0357	0.0355	0.0408	0.0366	0.0356	0.0314
2	0.0389	0.0349	0.0357	0.0405	0.0366	0.0343	0.0315
3	0.0394	0.035	0.0362	0.0416	0.0382	0.0348	0.0313
4	0.0389	0.0345	0.0357	0.0408	0.0379	0.0344	0.0314
5	0.0397	0.0351	0.036	0.0412	0.037	0.033	0.0316
6	0.0386	0.0356	0.0354	0.0409	0.0363	0.032	0.0313
7	0.04	0.036	0.0354	0.0408	0.0367	0.0333	0.0314
8	0.0391	0.0355	0.04	0.0408	0.0366	0.0334	0.0316
9	0.0393	0.0363	0.038	0.0412	0.0372	0.035	0.0315
10	0.0386	0.036	0.036	0.041	0.037	0.0332	0.0313
average	0.0392	0.0355	0.0364	0.0410	0.0370	0.0339	0.0314

Supplementary Fig. 4. Average weight of Ti₃C₂T_x-SCL and weight comparison with commercial transparent and color contact lenses.

Comment 3: The Ti₃C₂T_x -SCL with a high sensitivity of 12.52 mV mmHg⁻¹ can respond stably and significantly multiple times to pressure changes of different magnitudes. The authors are advised to give the smallest detection limit of Ti₃C₂T_x -SCL.

Answer: We really appreciate the reviewer's comments. The smallest detection limit is the lowest limit at which the sensor can accurately reflect what is being measured. The Ti₃C₂T_x -SCL was tested 80 times at different small pressures, and the output voltage changes between 0.025 mmHg and 0.05 mmHg appearing partially overlap, which have a large gap between 0.05 mmHg and 0.1 mmHg, as shown in Fig. S18a, b. Further statistical analysis indicates that the maximum relative error of the output voltage change at 0.025 mmHg is larger at 55.03%, while it is only 16.95% and 6.15% for 0.05 mmHg and 0.1 mmHg (Fig. S18c). The large relative error prevents the sensor from resolving pressures of 0.025 mmHg, so the smallest detection limit of Ti₃C₂T_x -SCL is 0.05 mmHg. To verify the consistency, 10 devices were tested at a pressure of 0.05 mmHg for 80 cycles, and Fig. S18d exhibits that the average measurement error is less than 3.6%, which demonstrates the Ti₃C₂T_x -SCL offers the smallest detection limit of 0.05 mmHg.

Supplementary Fig. 18. The output voltage variations (a) and the corresponding statistical distributions (b) of the $Ti_3C_2T_x$ -SCL at 0.025, 0.05 and 0.1 mmHg, respectively. c, Relative errors in the output voltage changes of the $Ti_3C_2T_x$ -SCL at 0.025, 0.05 and 0.1 mmHg. d, The average measurement error of the output voltage change obtained from 10 devices at a pressure of 0.05 mmHg.

Comment 4: In this paper, the authors prepared a neuroprosthetic contact lens for monitoring and feedback of intraocular pressure and further demonstrated the neuromorphic sensorimotor loop. The advantages of the neuroprosthetic contact lens and neuromorphic sensorimotor loop have been elaborated. However, the disadvantages of these devices and systems and corresponding improved methods should also be discussed.

Answer: We highly appreciate the reviewer's insightful questions. Currently, we use an external circuit board for signal acquisition and power supply, and the wires between the sensor and the board can be a bit inconvenient. Up to now, wearable IOP sensors can be divided into piezoresistive, inductive couple telemetry, microfluidic and structural color sensors according to its operating principle¹. The latter two do not require additional energy to drive the sensor, so they will not be discussed temporarily. Inductance-based IOP sensors use wireless coils for energy and signal transmission, which are more comfortable to wear, but have the disadvantages of low sensitivity, difficult processing, which limit their applications. While piezoresistive intraocular pressure sensors are currently mostly based on wires for energy supply and signal

transmission, with simple structure, high sensitivity and easier processing, this approach is widely adopted. In this work, the primary focus is on a sensorimotor system for neuroprosthetic contact lenses that enables instant intraocular pressure monitoring and motor feedback. In future research, more rational and comfortable ways of signal acquisition and energy supply will be explored, such as integrating processing chips and sensors into contact lenses, to provide solutions for further development into wearable IOP monitoring devices.

Comment 5: In recent years, some studies about retinomorph neurons and artificial reflex arcs based on neuromorphic devices for artificial vision or visual restoration have been reported. These works may have important applications in visual neuroprosthetics, thus the authors are advised to make a brief introduction to these artificial neurons and artificial reflex arcs. (Ref: Nature Communications 2023, 14, 7181; Nature Communications 2022, 13, 6760; Matter 2022, 5, 1578-1589; Materials Horizons 2023, 10, 5753-5762.)

Answer: Thanks for the reviewer's valuable comments. We have highlighted the retinomorph neurons and artificial reflex arcs as well as their applications in neuroprosthetics in the introduction parts, the suggested articles²⁻⁵ related to neuromorphic devices have been cited in the revised version as ref.16, ref.18-20.

References

- 1 Hi, Y. *et al.* Lab - on - a - Contact Lens: Recent Advances and Future Opportunities in Diagnostics and Therapeutics. *Adv. Mater.* **34**, 2108389 (2022).
- 2 Kim, S. *et al.* A biomimetic ocular prosthesis system: emulating autonomic pupil and corneal reflections. *Nat. Commun.* **13**, 6760 (2022).
- 3 Sun, L., Qu, S. & Xu, W. A retinomorph neuron for artificial vision and iris accommodation. *Mater. Horiz.* **10**, 5753–5762 (2023).
- 4 Qu, S. *et al.* An artificially-intelligent cornea with tactile sensation enables sensory expansion and interaction. *Nat. Commun.* **14**, 7181 (2023).
- 5 Gong, J. *et al.* An artificial visual nerve for mimicking pupil reflex. *Matter* **5**, 1578–1589 (2022).

Responses to Reviewer 2:

This work presented a neuroprosthetic contact lens enabled sensorimotor system, which consisted of a $Ti_3C_2T_x$ IOP strain sensor, a $Ti_3C_2T_x$ temperature sensor and an IOP monitoring/display system, to realize the point-of-care IOP monitoring and real-time display. With the IOP strain sensor, the system could convert the IOP information into an electrical signal, which was then corrected by the temperature sensor to obtain and display an accurate IOP value. Further, according to the level of IOP, the system could generate corresponding electrical signals to stimulate the neural center, and eventually formed a closed-loop of IOP signal generation-nerve-perception-motor activity. Finally, the authors demonstrated the good biocompatibility of the designed neuroprosthetic contact lenses through rabbit eye experiments, and verified the feasibility of the closed-loop IOP neurosensory activity through rat experiments. Based on the unique system design and the compelling IOP monitoring results, this manuscript is acceptable after major revision. The following comments need to be addressed before further consideration.

Answer: We appreciate your thoughtful assessment and comments on our manuscript. According to your suggestions, we have thoroughly reviewed the results and addressed the concerns you have raised. Your constructive suggestion is highly valued, and we have made the necessary revisions to improve the quality of our manuscript. Thank you again for your time and insights. Corresponding revisions and point-to-point responses are listed as following:

Comment 1: Normally, IOP would fluctuate cyclically due to a variety of factors such as day and night changes, emotional changes, and eye diseases. Accurately obtaining the upward and downward trends of IOP is necessary for the health monitoring of patients with eye diseases. From Figure 3b and Supplementary Figure 15a, it can be seen that under the same IOP value, there is a certain degree of difference between the output voltage of the $Ti_3C_2T_x$ IOP strain sensor in the rising phases and the falling phases. This means that the sensor is unable to accurately obtain IOP values based on a single-point test alone when the rising and falling trends are unknown, which is unacceptable for IOP monitoring in patients with eye diseases.

Answer: We thank the Reviewer for this question. We built a test platform to evaluate the performance of IOP sensors, including a simulated eyeball, a pressure regulation system, and a sensor signal test system, as shown in Fig. 3a and Fig. S11. The eyeball wall consists of a stainless steel chamber designed to mimic the sclera and a bionic cornea that enables the monitoring of pressure and temperature, while the contents are anhydrous ethanol and connected to an infusion bag that hangs from a motorized sliding table. The motorized sliding table can be programmed to control pressure changes within the bionic eye through the pressure adjustment system, whilst a manometer attached to the other end of the stainless steel chamber measures the pressure on the cornea in real time. In addition, the source meter in the pressure feedback system is connected to the sensor on the bionic cornea, where one channel provides a constant current to the sensor and the other channel is used to detect the output voltage value in response to the intraocular pressure.

To apply stepped pressure to the bionic eye, a program was written to stop the motorized sliding table for 10 s (the maximum dwell time allowed by the program) for each 6.25 mmHg change in pressure. The residence time of the motorized sliding table is too short to be sufficient for the anhydrous ethanol to reach equilibrium, leading to a certain degree of difference in pressure during rise and fall. We retested the static response of the single $Ti_3C_2T_x$ -SCL several times, and found our IOP sensors could capture the trends during rise and fall and reach balance, as shown in Fig. S17a and Fig. S17b (modified Figure 3b), which clearly displayed that the pressure and output voltage trends are the same during the rise and fall and finally become stable, allowing the $Ti_3C_2T_x$ IOP strain sensor to accurately obtain an IOP value. Moreover, to verify the repeatability and uniformity of the IOP sensors, the static response of three separate $Ti_3C_2T_x$ -SCL was also carried out (Fig. S17c and Fig. S17d), which shows negligible changes in pressure and output voltage, suggesting that the $Ti_3C_2T_x$ -SCL could be used for IOP monitoring.

Fig. 3a. The schematic of a simulated IOP testing platform with three modules: pressure control, device testing and data extraction. The inset is a photograph of the bionic eyeball.

Supplementary Fig. 15. Digital photograph of a simulated IOP testing platform containing bionic eyeball, motorized sliding table with program control system, manometer and source meter.

Fig. 3b. Static response of the $Ti_3C_2T_x$ -SCL with pressure change in steps of 6.25 mmHg.

Supplementary Fig. 17. Pressure change and static response of single $Ti_3C_2T_x$ -SCL and three separate devices increasing from 0 mmHg to 50 mmHg and then decreasing to 0 mmHg.

Comment 2: The neuroprosthetic contact lens enabled sensorimotor system, as a point-of-care IOP monitoring and real-time display system, needs to consider the signal transmission and power supply of the $Ti_3C_2T_x$ -SCL. Considering the wearing comfort and practicality in daily life, there are many excellent works that use wireless transmission or self-powered forms to solve the signal acquisition and power supply problems of IOP sensors. In this work, how was the issue of signal acquisition and power supply addressed? How does this affect the further development of this system into a wearable IOP monitoring system?

Answer: We really appreciate the reviewer's professional questions. In this work, we use an external circuit board for signal acquisition and power supply, and the wires between the sensor and the board can be a bit inconvenient. Up to now, wearable IOP sensors can be divided into piezoresistive, inductive couple telemetry, microfluidic and structural color sensors according to its operating principle¹. The latter two do not require additional energy to drive the sensor, so they will not be discussed temporarily. Inductance-based IOP sensors use wireless coils for energy and signal transmission, which are more comfortable to wear, but have the disadvantages of low sensitivity, difficult processing, which limit their applications. While piezoresistive intraocular pressure sensors are currently mostly based on wires for energy supply and signal transmission, with simple structure, high sensitivity and easier processing, this approach is widely adopted. In this work, the primary focus is on a sensorimotor system for neuroprosthetic contact lenses that enables instant intraocular pressure monitoring and motor feedback. In future research, more rational and comfortable ways of signal acquisition and energy supply will be explored, such as integrating processing chips and sensors into contact lenses, to provide solutions for further development into wearable IOP monitoring devices.

Comment 3: This work verified that the detection range (0-50 mmHg) and detection sensitivity (12.52 mV/mmHg-1) of the IOP strain sensor could meet the needs of both the normal population and the ocular disease patient population. However, the accuracy and measurement error of the IOP stress sensor is not statistically analyzed in the manuscript. In addition, what is the effective resolution of IOP strain sensors during the IOP detection? The authors should conduct further investigation on these aspects.

Answer: We highly appreciate the reviewer's responsible attitude and detailed guidance towards a better manuscript. We performed more detailed testing and statistical analysis of the devices, as shown below:

The measurement error of the sensor is mainly divided into systematic error and random error, where the systematic error is caused by the inherent characteristics of the sensor itself or imperfections in the manufacturing process, and the random error is caused by external environmental factors and changes in the measurement conditions, which are calculated as follows:

$$E = |V_m - V_s| \quad (2)$$

Where E is the error of the sensor and V_m and V_s are the measured and standard values at the same pressure.

To obtain the measurement errors of the IOP stress sensor, 8 devices were randomly selected and tested at pressures of 0.05, 0.1, 0.15, 0.2 and 0.25 mmHg, and 80 points were statistically analyzed at each pressure, as reported in Table S4. Fig. S21a, b shows that the measurement errors of all devices are in good agreement at the same pressure. Fig. S21c, d further indicates that the average measurement error of the 8 devices is 0.030 mV at a pressure of 0.05 mmHg, 0.030 mV of 0.1 mmHg, 0.028

mV of 0.15 mmHg, 0.026 mV of 0.20 mmHg, 0.031 mV of 0.25 mmHg, and that the average measurement error of IOP strain sensor is 0.029 mV.

Supplementary Fig. 21. Average output voltage variations and average measurement errors for each IOP stress sensors at pressures of 0.05, 0.1, 0.15, 0.2, and 0.25 mmHg, respectively (a) as well as statistical distributions of the measurement errors for each device at a pressure of 0.2 mmHg (b). Average output voltage variations and average measurement errors of the IOP stress sensors (c) as well as the statistical distribution of the average measurement error at all pressures (d) at pressures of 0.05, 0.1, 0.15, 0.2, and 0.25 mmHg, respectively.

Supplementary Table 4: Maximum measurement error, minimum measurement error and average measurement error at 0.05, 0.1, 0.15, 0.2 and 0.25 mmHg (Unit: mV).

Decive		0.05 mmHg	0.1 mmHg	0.15 mmHg	0.2 mmHg	0.25 mmHg	average
1	max	0.145	0.084	0.069	0.084	0.097	0.096
	min	-0.098	-0.114	-0.064	-0.083	-0.084	-0.089
	average	0.036	0.028	0.028	0.026	0.029	0.029
2	max	0.088	0.073	0.086	0.092	0.096	0.087
	min	-0.079	-0.080	-0.074	-0.090	-0.083	-0.081
	average	0.029	0.028	0.031	0.025	0.029	0.028
3	max	0.123	0.087	0.061	0.076	0.127	0.095
	min	-0.099	-0.132	-0.082	-0.111	-0.121	-0.109
	average	0.034	0.035	0.025	0.025	0.033	0.030
4	max	0.110	0.125	0.111	0.063	0.084	0.098
	min	-0.098	-0.075	-0.103	-0.066	-0.083	-0.085

	average	0.032	0.026	0.028	0.024	0.033	0.028
5	max	0.110	0.086	0.082	0.073	0.12	0.094
	min	-0.114	-0.078	-0.085	-0.087	-0.115	-0.096
	average	0.028	0.031	0.027	0.033	0.036	0.031
6	max	0.085	0.071	0.092	0.059	0.12046	0.086
	min	-0.060	-0.063	-0.087	-0.072	-0.0965	-0.076
	average	0.022	0.028	0.029	0.028	0.03406	0.028
7	max	0.075	0.088	0.078	0.080	0.0694	0.078
	min	-0.068	-0.112	-0.070	-0.046	-0.07126	-0.074
	average	0.025	0.032	0.029	0.022	0.026	0.027
8	max	0.101	0.068	0.080	0.076	0.06679	0.078
	min	-0.103	-0.078	-0.080	-0.081	-0.05957	-0.080
	average	0.038	0.030	0.026	0.029	0.02519	0.030
average		0.030	0.030	0.028	0.026	0.031	0.029

Accuracy is the difference between the measurement result and the standard value, and is calculated by the following formula:

$$A = \frac{V_m}{V_s} \quad (3)$$

where A represents the accuracy of the device, V_m is the average deviation and V_s is the standard value.

We performed 80 tests on each of the 15 devices at a pressure of 0.2 mmHg and calculated the corresponding standard values, average deviations and accuracies, as listed in Table S3 and Fig. S20, showing the accuracy of the $Ti_3C_2T_x$ -SCL at 0.2 mmHg pressure of 1.075% as a result of statistical analysis.

Supplementary Table 3: Standard value, average deviation and accuracy of the $Ti_3C_2T_x$ -SCL at 0.2 mmHg.

Decive	Standard value (mV)	Average variation (mV)	Accuracy (%)
1	2.715	0.026	0.955
2	2.731	0.025	0.899
3	2.749	0.025	0.902
4	2.76	0.024	0.856
5	2.797	0.033	1.173
6	2.821	0.028	0.999
7	2.790	0.022	0.774

8	2.774	0.033	1.189
9	2.746	0.039	1.408
10	2.705	0.030	1.108
11	2.675	0.030	1.111
12	2.675	0.029	1.083
13	2.649	0.028	1.071
14	2.638	0.040	1.517
15	2.583	0.028	1.071
			
Average value	2.721	0.029	1.075

Supplementary Fig. 20. Statistical analysis of the $\text{Ti}_3\text{C}_2\text{T}_x$ -SCL accuracy at 0.2 mmHg.

Effective resolution is an important parameter for IOP strain sensors as it is the smallest change that can be detected by the sensor. The $\text{Ti}_3\text{C}_2\text{T}_x$ -SCL was tested 80 times at different small pressures, where the output voltage changes at 0.025 mmHg and 0.05 mmHg appears to partially overlap, and there is a large gap between the output voltage changes at 0.05 mmHg and 0.1 mmHg, as shown in Fig. S18a, b, which indicated that the device cannot discriminate between the 0.025 mmHg of pressure change, but can discriminate between the 0.05 mmHg of pressure change. In order to verify the homogeneity of the $\text{Ti}_3\text{C}_2\text{T}_x$ -SCL, we performed pressure tests on 6 random devices at intervals of 0.05 mmHg, respectively, and Fig. S19a-f displays that there is a very significant difference in the output voltage variations of each device at different pressures. The statistical analysis of the output voltage change at each pressure (0.05, 0.1, 0.15, 0.2 and 0.25 mmHg) in Fig. S19g further indicates that the effective resolution of the $\text{Ti}_3\text{C}_2\text{T}_x$ -SCL is 0.05 mmHg.

Supplementary Fig. 18. The output voltage variations (a) of the $\text{Ti}_3\text{C}_2\text{T}_x$ -SCL at 0.025, 0.05 and 0.1 mmHg, respectively, and the corresponding statistical distributions (b).

Supplementary Fig. 19. Output voltage variations (a-f) and statistical distribution (g) of the $Ti_3C_2T_x$ -SCL at 0.05, 0.1, 0.15, 0.2 and 0.25 mmHg, respectively.

Comment 4: During the sensing process of IOP strain sensor and temperature sensor, is there a time lag (i.e., a hysteresis in the sensor's response) in the detection of the IOP signal and the temperature signal? If so, does the time lag differ between IOP detection and temperature detection? And how would this difference affect the accuracy of the final IOP results?

Answer: Many thanks for your comments. To examine the response of the IOP sensor, cycling tests were performed at 1x speed over the range of IOP changes from 0-21 mmHg. Fig. S22 shows that the output voltage change is synchronized with the pressure change without significant delay, and the amplitude of the voltage change (ΔV) is relatively stable, indicating that the sensor can respond accurately to IOP fluctuations. The dynamic response of the temperature sensor was also measured by slowly increasing the temperature at a rate of 1°C min^{-1} , as shown in Fig. S30, it can be seen that the dynamic current response is consistent with the increase of the temperature, indicating that there is almost no time lag in the detection of the temperature signal. As a result, the $\text{Ti}_3\text{C}_2\text{T}_x$ -SCL simultaneously senses both temperature and pressure bimodal signals and outputs accurate IOP results over a wide temperature range.

Supplementary Fig. 22. Cycling test at 1x speed in the range of 0-21 mmHg.

Supplementary Fig. 30. Current dynamic response of temperature sensors during slow temperature rise.

Comment 5: How does the neuroprosthetic contact lens enabled sensorimotor system connect with the somatosensory cortex, motor cortex, and sciatic nerve to generate electrical stimulation during in vivo experiments in rats? And how were the

corresponding potential signals collected at the different positions? The authors should describe the corresponding parts of the article in more detail.

Answer: Many thanks for your insightful question about the details of the rat experiment. This work focuses on remodeling the sensorimotor circuits based on the eyeball-somatosensory cortex-motor cortex-sciatic nerve by stimulating rats with a neuroprosthetic contact lens, and collecting the corresponding potential signals using a PowerLab data acquisition device and LabChart analysis system. In this system (Fig. 5b and Fig. S38), the Ti_3C_2Tx -SCL simultaneously senses both temperature and pressure bimodal signals and collects them via an analog-to-digital converter (ADC) integrated within the microcontroller unit (MCU). The DAC module built into the MUC regulates the output current of the voltage-controlled current source based on the IOP level and then switches the current through analog switches to produce a pulsed current stimulation signal.

The transmission of nerve impulses throughout the nervous system from the eye to the limb during IOP abnormalities was investigated hierarchically in a live rat model. When induced a high IOP via saline injection into rat eyes, there were no obvious changes of potentials in the somatosensory cortex based on the result of Fig. S34, which suggests the weak neural oscillations between the eye and the brain, indicating the existence of the individual variations in pain perception and sympathetic responses. Therefore, constructing an association between high IOP and somatosensory cortex is of great significance. In this process, a simulated eyeball was used to create an IOP abnormality and stimulate the rat's somatosensory cortex by generating impulse signals through the neuroprosthetic contact lens, recording consistent potential changes with the stimulation signal in this cortex (Fig. S35), suggesting that our device can modulate neural oscillations between the eyeball and brain, allowing rats to feel the IOP change. Meanwhile, potential changes corresponding to the stimulus signals were similarly captured in the motor cortex and gastrocnemius muscle when neuroprosthetic contact lens detected IOP abnormalities as shown in the main text Fig. 5a III and Fig. S36, 37, respectively, confirming the feedback of sensory signals by the rat organism. In the whole nervous system, only the neural oscillations between the eyeballs to the somatosensory cortex are absent or weak, so we simulated different levels of IOP grades and performed experiments in rats. Potential changes of varying intensity were detected in the somatosensory cortex (Fig. S39), suggesting that neuroprosthetic contact lenses may be able to evoke the perception of intraocular pressure in the brain and thereby remodel the neuromorphic sensorimotor circuitry. The corresponding description has been added in the revised manuscript.

Fig. 5b, System block diagram of the pressure and temperature signals processing step including signal acquisition, processing, control, communication and display.

Supplementary Fig. 38. Circuit boards for data acquisition, processing and generation of corresponding stimulus signals.

Supplementary Fig. 34. Potential signals from the somatosensory cortex collected during intraocular injection of saline into the rat eye.

Supplementary Fig. 35. Stimulation of the somatosensory cortex by impulse signals generated by the neuroprosthetic contact lens in response to abnormal intraocular pressure in the rat and the corresponding potential signals collected in the somatosensory cortex, where panel b is a detailed view of a single pulse cluster in panel a.

Fig. 5aIII. The stimulation signals to the somatosensory cortex and the corresponding potential signals collected in the motor cortex during abnormal intraocular pressure in rats.

Supplementary Fig. 36. Stimulation of the somatosensory cortex by impulse signals generated by the neuroprosthetic contact lens in response to abnormal intraocular pressure in the rat and the corresponding potential signals collected in the leg, where panel b is a detailed view of a single pulse cluster in panel a.

Supplementary Fig. 37. Stimulation of the motor cortex by impulse signals generated by the neuroprosthetic contact lens in response to abnormal intraocular pressure in the rat and the corresponding potential signals collected in the leg, where panel b is a detailed view of a single pulse cluster in panel a.

Supplementary Fig. 39. Different levels of the potential signals collected in the somatosensory cortex when the rats were in different IOP ranges.

Comment 6: In this work, a neuromorphic sensorimotor loop design was proposed (Figure 5a). However, in the subsequent actual experiment (Fig. 5c), it seems that the authors directly electrically stimulated the sciatic nerve through the circuit to realize the effect of controlling the movement of the leg, without the involvement of the somatosensory cortex and the motor cortex of the brain? Could the authors please

provide a reasonable explanation for the discrepancy between the description of the neuromorphic sensorimotor loop and the actual experimental test?

Answer: Thank you very much for your professional comment. The function of the neuromorphic sensorimotor loop was demonstrated as following process: When a neuroprosthetic contact lens detected abnormal IOP in a simulated eyeball that will generate pulsed signals to stimulate in the rat somatosensory cortex, graded potential changes were detected in the somatosensory cortex (Fig. S39), suggesting that our device could modulate neural oscillations between the eyeball and the brain to allow rats to perceive graded IOP changes. The whole process was provided in video S3, which obviously displays the rat leg twitching under the control of the somatosensory cortex when IOP levels were abnormal.

However, the graded potential changes in the somatosensory cortex led to a small leg twitching angle, the angular changes between different IOP levels could not be well observed at a small twitching angle, therefore we added a supplemented test by direct stimulation of the sciatic nerve to amplify the effect of leg movements. The experiment simulated different eye conditions by controlling the pressure within the bionic eye, including low intraocular pressure (0-9 mmHg), healthy intraocular pressure status (10-21 mmHg), and high intraocular pressure (22-30, 31-40, and 41-50 mmHg). As shown in Fig. 5c and Video S4, there is no stress response in the leg of rat under normal IOP, but the leg exhibits a slight angle of flexion, approximately 22° when the intraocular pressure is below normal. In contrast, when the eye is under high IOP conditions, the leg exhibits a greater angle of flexion, which increases with increasing IOP values, reaching 44° at 22-30 mmHg, 62° at 31-40 mmHg, and 76° at 41-50 mmHg. The signal intensities corresponding to different IOP levels acquired on the gastrocnemius muscle demonstrate that the neuroprosthetic contact lens based on the $\text{Ti}_3\text{C}_2\text{T}_x$ MXene can provide real-time and graded sensory feedback (Fig. 5d). The full description has been added in the revised manuscript.

Supplementary Fig. 39. Different levels of the potential signals collected in the somatosensory cortex when the rats were in different IOP ranges.

Fig. 5. c, Photographs of leg twitch responses in a live rat model and real-time IOP display stimulated at various simulated IOP levels. d, Signals acquired in the gastrocnemius muscle while stimulated at various IOP levels in a living rat model.

Comment 7: Figure 5d was confusing, and it should be clearly labeled and explained what the different colored curves represent. In addition, there is no difference in amplitude and frequency between the electrical stimulation signals (black curves) in various IOP ranges, which was obviously not in accordance with the actual situation. The authors should make a clear revision and explanation of Figure 5d.

Answer: We really appreciate the reviewer's professional questions. In Fig. 5d, the red, purple, pink, green, and blue curves correspond sequentially to data collected in the gastrocnemius muscle when IOP is in the range of 0-9 mmHg, 10-21 mmHg, 22-30 mmHg, 31-40 mmHg, and 41-50 mmHg. The colored curves in the dashed box in the second column are locally magnified graphs of the signals in the first column. The gray curve collected by the circuit boards only reveals the stimulus waveform and frequency verifies whether the existence or nonexistence of stimulation signal in the neuromorphic contact lens at different IOP levels. We have explicitly revised and explained this section in the manuscript.

Fig. 5. d, Signals acquired in the gastrocnemius muscle while stimulated at various IOP levels in a living rat model (The red, purple, pink, green, and blue curves correspond sequentially to data

collected in the gastrocnemius muscle when IOP is in the range of 0-9 mmHg, 10-21 mmHg, 22-30 mmHg, 31-40 mmHg, and 41-50 mmHg. The colored curves in the dashed box in the second column are locally magnified graphs of the signals in the first column. The gray curve shows the stimulation signal of the NCL.

Comment 8: This manuscript consisted of two main parts: one was the preparation and performance testing of an IOP strain sensor with temperature correction; the other was a neuromorphic sensorimotor loop for IOP monitoring through stimulation of the neural center. In this work, after obtaining IOP information from the IOP strain sensor and estimating whether IOP levels were abnormal, the authors used an external circuit system to electrically stimulate the somatosensory cortex, sciatic nerve, and other parts of the brain for neuromorphic sensorimotor feedback. While the two parts of the work were convincingly presented and achieved good results, the whole work was more like two separate pieces of work cobbled together. The authors should fully explain the intrinsic connection between the two pieces of work to better demonstrate the integrity and consistency of the manuscript.

Answer: Thank you for your comment. In this work, we are trying to mimic natural IOP stimulated-nerve-induced motor activities and construct an association from high IOP to neurons. When induced a high IOP via saline injection into rat eyes, there were no obvious changes of potentials in the somatosensory cortex based on the result of Fig. S34, which suggests the weak neural oscillations between the eye and the brain, indicating the existence of the individual variations in pain perception and sympathetic responses. Therefore, constructing an association between high IOP and somatosensory cortex is of great significance. In detail process, as shown in Fig. 5b, when applied an unknown IOP, the circuit system simultaneously processes the temperature and pressure signals to obtain the temperature-corrected accurate IOP level and decode the IOP status. If the IOP is abnormal state, the somatosensory cortex is stimulated by a pulsed current, which will delivered to the motor cortex, leading to the rat leg twitching (Video S3).

Supplementary Fig. 34. Potential signals from the somatosensory cortex collected during intraocular injection of saline into the rat eye.

Fig. 5. b, System block diagram of the pressure and temperature signals processing step including signal acquisition, processing, control, communication and display.

Comment 9: The authors are suggested to improve the English of the manuscript. It is better to be edited by native English speakers.

Answer: We highly appreciate your comments. Our manuscript has been reviewed by a native English speaker, and revised to improve readability

REVIEWERS' COMMENTS

Reviewer #1 (Remarks to the Author):

The authors have provided detailed answers to the reviewers' questions in the revised manuscript. The revised manuscript is complete and it is recommended to be published in this journal.

Reviewer #3 (Remarks to the Author):

The authors have addressed the comments. The paper is acceptable now.